# Neural Network-Based Dual-Cylinder Synchronous Control of a Multi-Link Erection Mechanism

**Weilin Zhu [1], Yaowen Ge [1], Wenxiang Deng [1,*], Lan Li [2], Xiangxin Liu [2], Jialin Zhang [2] and Jianyong Yao [1]**

1   School of Mechanical Engineering, Nanjing University of Science and Technology, Nanjing 210094, China
2   Beijing Institute of Space Launch Technology, Beijing 100081, China
*   Correspondence: wxdeng@njust.edu.cn; Tel.: +86-025-8431-5125

**Abstract:** A dual-cylinder erection mechanism, in which two telescopic cylinders physically connect to a load, is a nonlinear system with model uncertainties and coupled dynamics. In this paper, a novel synchronous control algorithm with thrust-allocation law is proposed for eliminating the excessive internal forces caused by the unbalanced rotation and lateral moments during the erection process. With regulated internal forces, the "pull and drag" issue is attenuated and better synchronization performance is attained. For improved tracking accuracy, the inter-stage collision dynamics of the telescopic cylinder are considered for model compensation to enhance stage-changing and in-position performance. A radial basis function (RBF) neural network is utilized to estimate the model uncertainties and external disturbances, which alleviates reliance upon the accuracy of a system model for controller implementation. As a result, theoretical analysis revealed that the semi-global asymptotic stability and synchronized motion performance with decreased internal forces can be achieved via the presented synchronous controller with thrust-allocation strategy. Contrasting simulations were implemented on a multi-link erection mechanism and the results confirmed the superiority and effectiveness of the proposed synchronous control algorithm.

**Keywords:** dual-cylinder erection mechanism; synchronous control; thrust allocation; inter stage collision; neural network

## 1. Introduction

Since the superiorities of large power-to-weight ratio, large force/torque supply, high redundancy and high concordance, dual-drive actuators have extensive applications in diverse industrial fields, such as gantry cranes, lifting systems and robotic manipulators [1–5]. As studied in this paper, a large-scale erection mechanism is driven by two telescopic cylinders in a collaborative manner, which uses a multi-link mechanism to transmit power. A physical connection of the actuators to the erection load is necessary for joint thrust generation. Thus, the two telescopic cylinders might interfere with each other, from which emerges the "pull and drag" problem [6,7], leading to load deformation and destruction. Nevertheless, it is essential to ensure synchronous motions of dual cylinders without excessive internal forces for a stable erection. In addition, the electro-hydraulic servo-erection system under study is subjected to many nonlinearities [8–10] including mechanism nonlinearities arising from the conversion between the displacement and rotation, distinct dynamics of the asymmetric two chambers of the telescopic cylinder, and pressure and flow nonlinearities of the hydraulic system. Furthermore, the electro-hydraulic erection mechanism generally suffers from various model uncertainties, such as friction, flow leakage and external disturbances [11–13]. Therefore, the development of a high-performance control approach has significant theoretical and practical implications for the electro-hydraulic erection system.

So far, scholars have investigated several synchronous control algorithms of multiple actuators to follow the same motion trajectory. Generally, the commonly used synchronous

control strategies [14–16] can be classified into three categories: (a) parallel synchronization method or synchronized master command generator, in which the multiple actuators follow the respective control loops severally, (b) master-slave synchronous control scheme, in which the instruction to the slave actuator is generated from the master actuator who follows the ideal trajectory, and (c) cross-coupling synchronous control scheme, in which a synchronous compensator is established on the basis of position or velocity discrepancies from dual actuators [17]. The first two designs are simple to implement but have intrinsic performance restrictions due to the non-sharing of feedback information and a time delay of control signals. The third design adds position or velocity feedback and discrepancies to the synchronized master command generator, and thus a closed-loop control framework is formed to achieve better motion synchronization. It is worth noting that there are radical distinctions between the synchronous control of the dual-cylinder erection system investigated in this paper and the synchronous or coordinated control of two individual actuators for trajectory tracking missions. The dynamics of dual-cylinder drives in the synchronous control design considered in this paper are fully coupled, which is different from the basically decoupled dynamics of the coordinated motion control. In particular, owing to the hinges between the two cylinders, linkage mechanism and erection load, physical constraints restrict dual-cylinder relative motion or mechanically synchronize the motions of each other. if the control performance of the dual cylinders is not highly synchronized, excessive internal forces caused by unbalanced rotation and lateral moments may result in performance deterioration, load deformation and even damage to the system components [7]. Aware of these potential adverse effects, the synchronous control strategy is not only required to synchronize motions but also to avoid excessive internal forces for erection stability. Nevertheless, the issue of appropriately handling internal forces of such synchronization systems with coupled dynamics has been neglected in most synchronous control strategies.

As for the advanced motion control algorithms, a sliding mode control strategy called robust integral of the sign of the error (RISE), first presented in [18], is utilized to handle matched uncertainty of nonlinear systems. This controller was integrated into an adaptive backstepping framework in [19–22] to pledge asymptotic tracking performance in the presence of various uncertainties. In [23], Yao et al. used an adaptive RISE controller to handle parametric uncertainty and unmodeled disturbance, which can acquire excellent transients and steady-state tracking performance. However, high-gain or high-frequency feedback was usually adopted to eliminate severe unknown dynamics, which may deteriorate the control precision. The radial basis function neural network (RBFNN) is a universal feedforward approximator widely utilized in the intelligent control field. Due to less reliance on an accurate system model and fast convergence speed, it has been broadly applied to approximate and compensate unknown dynamics and indeed improve the tracking performance of mechanical and hydraulic servo systems in [24–27]. The RBF neural network was combined with a sliding mode control method to approximate model uncertainties and excellent performance was acquired in [24–26]. The RBF neural network was validly integrated with a continuous RISE feedback term to deal with the uncertain dynamics and external disturbance and attain asymptotic stability in [27]. Motivated by the above observation, the characteristics of the RBF neural network, such as simple architecture, accelerating learning speed, and avoiding the local minimum issue, which are all crucial factors in real applications, make it suitable for experiment implementation. Hence, it is worth believing that the model uncertainties and disturbance of the real erection system can be well approximated and compensated through the RBFNN.

However, the combination and design of a synchronous control strategy and the RBFNN for the dual-cylinder erection mechanism is still rare in the existing literature. In this paper, a novel synchronous control strategy with thrust allocation law is proposed to address excessive internal forces generated from complex coupling dynamics mentioned above. Distinguished from the existing sheer synchronous or coordinated motion control schemes, the presented synchronous control strategy focuses not only on synchronization

performance of the dual cylinders but also the regulation of internal forces for erection stability. Specifically, in step 1, the dual-cylinder kinematics model with thrust-allocation law is derived with the knowledge of unbalanced rotational dynamics for subsequent synchronous controller design. To avert deterioration of stage-changing and in-position tracking performance, the inter-stage collision dynamics of the telescopic cylinder are established for model-based compensation. A dynamic model of the erection system is then derived, considering mechanism kinematics, hydraulic dynamics, and external disturbances. An RBFNN is utilized to approximate the unknown dynamics and a RISE control law is synthesized to address the residual uncertainties. Subsequently, in step 2, the virtual-control input thrusts of the two cylinders are derived on the basis of total thrust virtual-control input generated in step 1 via the thrust-allocation strategy, and then the actual control inputs for the dual-cylinder erection system can be obtained. The semi-global asymptotic stability is acquired via Lyapunov analysis. Comparative simulations of the dual-cylinder erection mechanism demonstrate the effectiveness of the RBFNN approximator and the presented synchronous control strategy.

The contributions of this manuscript mainly include the following aspects: (1) A novel synchronous control strategy with thrust allocation law is proposed to deal with the "pull and drag" issue and to regulate excessive internal forces caused by coupled rotational dynamics, which is distinguished from the existing sheer motion synchronization schemes. (2) On the basis of an improved damping model, the inter-stage collision dynamics of the telescopic cylinder are constructed for model compensation to attain better stage-changing and in-position tracking performance. (3) An RBFNN is incorporated into the synchronous control strategy design for unknown dynamics approximation and compensation. (4) Comparative simulations of the dual-cylinder erection mechanism reveal the superiority of the presented synchronous control strategy.

This paper is arranged as follows: Section 2 gives the system modeling of the dual-cylinder erection mechanism. Section 3 presents the neural network-based synchronous control strategy design procedure and its stability analysis. Comparative simulation results are given in Section 4. The conclusions are contained in Section 5.

## 2. System Modeling

The large-scale erection system under research is depicted in Figure 1. Multi-link mechanisms are mounted symmetrically on both sides of the erection load, each of which consists of five hinge points. The erection mechanism is driven by two telescopic cylinders. The aim is to enable the erection load to track the desired erection trajectories as closely as possible and decrease the synchronous errors of two cylinders simultaneously.

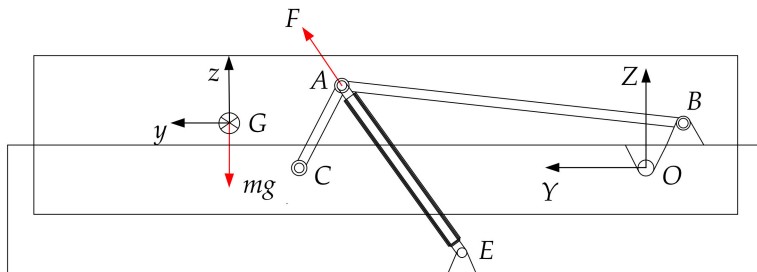

**Figure 1.** Schematic diagram of the erection system.

### 2.1. Dynamics of Multi-Link Erection Mechanism

Using the motion vector equations of a multi-link mechanism, the following expression can be obtained based on Figure 2.

$$\varphi = 2\arctan\frac{b + \sqrt{a^2 + b^2 - c^2}}{a - c} \tag{1}$$

where $a = L_4 - L_1 \cos(\beta_0 - \theta)$, $b = -L_1 \sin(\beta_0 - \theta)$, $c = 0.5(a^2 + b^2 - L_2^2 + L_3^2)/L_3$ are defined as auxiliary variables to solve $\angle ABH$; $L_1$, $L_2$, $L_3$ and $L_4$ denote the length of linkage $\overline{CO}$, $\overline{AC}$, $\overline{AB}$ and $\overline{BO}$ respectively; $\beta_0$ denotes the value of $\angle BOC$; $\theta$ denotes the erection angle.

The axial displacement of the telescopic cylinder $\overline{AE}$ can be obtained based on the cosine theorem as

$$x_p = \sqrt{L_3^2 + L_5^2 - 2L_3 L_5 \cos(\pi - \alpha - \varphi)} - L_6 \tag{2}$$

where $L_5$ denotes the distance from $B$ to $E$; $L_6$ is the initial length of the telescopic cylinder; and $\alpha$ denotes the value of $\angle OBE$.

According to the moment equilibrium, the dynamics of the erection system can be given as

$$J\ddot{\theta} = \tau(\theta)(F_1 + F_2) - mgL_8 \cos(\beta_2 + \theta) - B\dot{\theta} - A_f S_f(\dot{\theta}) - d(t) \tag{3}$$

where $J$ represents the rotational inertia of the load relative to $O$; $\tau(\theta) = \partial x_p / \partial \theta$ represents the erection arm of force; $F_1$ and $F_2$ are the thrust forces of the two telescopic cylinders severally; $m$ is the mass of erection load; $L_8$ denotes the distance from $G$ to $O$; $\beta_2$ represents the value of $\angle COG$; $B$ represents the viscous friction coefficient; $A_f S_f$ represents the approximated Coulomb friction, in which $A_f$ is the Coulomb friction amplitude and $S_f$ is a known shape function; $d$ denotes unknown dynamics including friction, external disturbances and unmodeled dynamics.

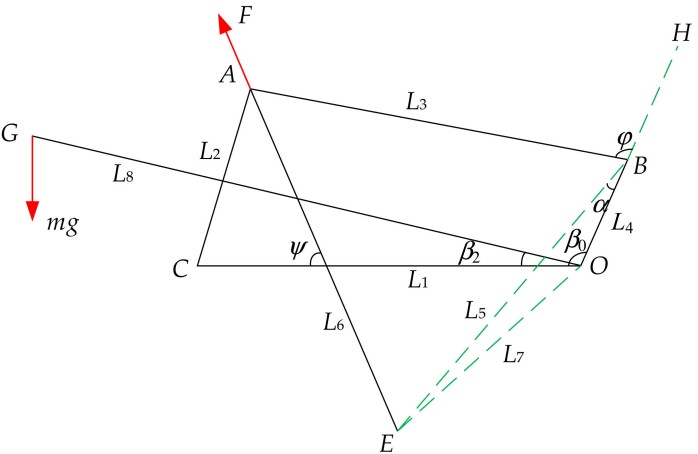

**Figure 2.** Schematic diagram of the multi-link mechanism.

### 2.2. Dynamics of Inter-Stage Collision

During the extension process of the telescopic cylinder, the position is limited by the contact collision between the cylinder shell and piston rod, which has a great effect on the dynamic characteristics of the erection process. In order to calculate the collision force, the collision process is decomposed into four states: contact-deformation-recovery-detachment [28]. Deformation is generally limited in the neighborhood of the contact area. The spring contact force is determined based on the Hertz contact law, and the energy loss is considered through a damper parallel to the spring. The cylinder motion model is depicted in Figure 3.

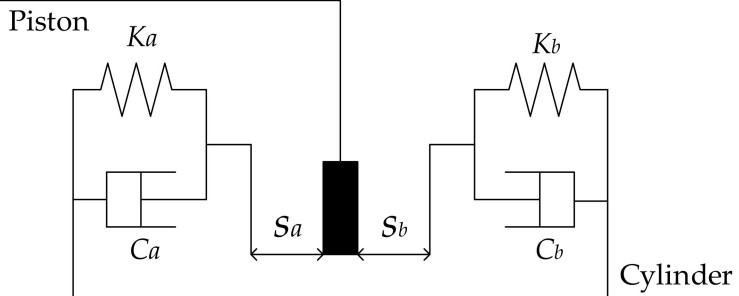

**Figure 3.** Schematic diagram of the cylinder motion model.

The collision force model considering damping loss was first proposed by Kelvin-Voigt. The formula is as follows

$$F_N = \begin{cases} K_a \delta^n + C_a \dot{\delta} & x \leq S_a \\ 0 & S_a < x < S_b \\ K_b \delta^n + C_b \dot{\delta} & x \geq S_b \end{cases} \tag{4}$$

where $K_a$ and $K_b$ denote effective spring stiffness; $C_a$ and $C_b$ denote effective damping coefficient; $\delta$ is normal penetration depth in contact position, and $\dot{\delta}$ is normal relative velocity of the contact point; $n$ is material-related nonlinear coefficient; $x$ denotes piston displacement; $S_a$ and $S_b$ are the lower and upper values of stroke, respectively.

As expressed in the formula, the energy loss of collision is described via a viscous damper with constant damping coefficient. Nevertheless, the contact force exists in the light of the above formula, which is not in accord with the actual state. Subsequently, a modified equivalent damping model with hysteresis factor is proposed, considering the relation between energy loss and deformation.

$$F_N = \begin{cases} K_a \delta^n + D\delta^n \dot{\delta} & x \leq S_a \\ 0 & S_a < x < S_b \\ K_b \delta^n + D\delta^n \dot{\delta} & x \geq S_b \end{cases} \tag{5}$$

where hysteresis damping factor $D$ is given as

$$D = \frac{3k(1 - C_r^2)e^{2(1-C_r)}}{4\dot{\delta}_0} \tag{6}$$

in which $C_r$ is Newton's recovery coefficient; $\dot{\delta}_0$ is the initial impact velocity.

The computed results of deformation and contact force are illustrated in Figures 4 and 5. There exist no deformation and contact forces at the inception phase, which is consistent with the real situation. As deformation rises to the maximum value, contact force reaches the maximum value. Subsequently, contact force reduces with deformation recovery. Since the instantaneous collision force is large, the effect on the erection process cannot be neglected as unmodeled uncertainties, which may degrade the transient control performance.

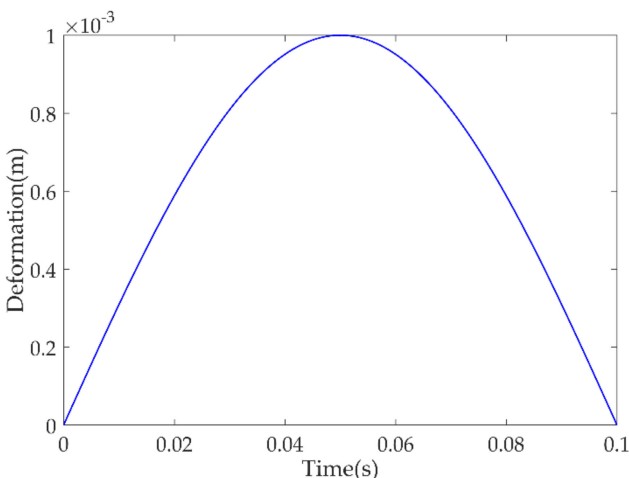

**Figure 4.** Schematic diagram of collision deformation.

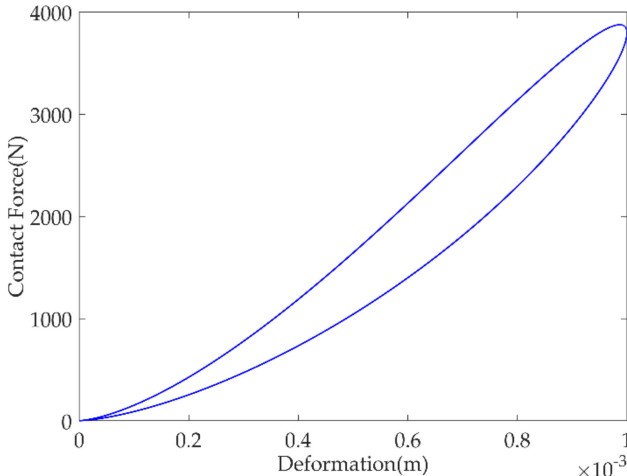

**Figure 5.** Contact force curve of collision.

### 2.3. Dual-Cylinder Kinematics with Thrust Allocation

Due to manufacturing and assembly, load eccentricity issues broadly exist in the practical mechanism. As shown in Figure 1, $OXYZ$ is defined as the fixed inertial frame with its origin at the rotation point, The X-axis is inward perpendicular to plane; the Y-axis is parallel to the ground; and the Z-axis is perpendicular to the ground. $G$ is defined as the centroid of the inertia load. $Gxyz$ is defined as a moving ideal coordinate frame at $G$ with the y-axis parallel to the inertia load and the z-axis perpendicular to the inertia load. $G\widetilde{x}\widetilde{y}\widetilde{z}$ is defined as a physically fixed coordinate frame to the inertia load at $G$, which revolves with the unbalanced rotation of the inertia load during the erection process.

As shown in Figure 6, let $D_1$ and $D_2$ denote the force point of the unbalanced rotation force. The positions of $D_1$, $D_2$ and $G$ are denoted as $D_1(x_1, y_1, z_1)$, $D_2(x_2, y_2, z_2)$ and $G(x_G, y_G, z_G)$ in $OXYZ$ respectively. $G\widetilde{x}'\widetilde{y}'$ is the projection of $G\widetilde{x}\widetilde{y}$ on plane $Gxy$ and $G\widetilde{x}'\widetilde{z}'$ is the projection of $G\widetilde{x}\widetilde{z}$ on plane $Gxz$. Let $\omega_1$ denote the rotation angle of $G\widetilde{x}'\widetilde{y}'$ relative to $Gxy$ and $\omega_2$ denote the rotation angle of $G\widetilde{x}'\widetilde{z}'$ relative to $Gxz$. Denote the distances

from the centroid $G$ to the two force points as $|GD_1| = l_1$ and $|GD_2| = l_2$, and the distance between the two force points as $|D_1 D_2| = l$, as seen from Figure 6,

$$
\begin{aligned}
x_1 &= x_G - l_1 \cos \omega_1 \approx x_G - l_1 \\
y_1 &= y_G - l_1 \sin \omega_1 \approx y_G - l_1 \omega_1 \\
z_1 &= z_G - l_1 \sin \omega_2 \approx z_G - l_1 \omega_2 \\
x_2 &= x_G + l_2 \cos \omega_1 \approx x_G + l_2 \\
y_2 &= y_G + l_2 \sin \omega_1 \approx y_G + l_2 \omega_1 \\
z_2 &= z_G + l_2 \sin \omega_2 \approx z_G + l_2 \omega_2
\end{aligned}
\tag{7}
$$

From which $x_G$, $\omega_1$, and $\omega_2$ can be obtained by

$$
x_G = \frac{l_2}{l} x_1 + \frac{l_1}{l} x_2, \, \omega_1 = \frac{1}{l}(y_2 - y_1), \, \omega_2 = \frac{1}{l}(z_2 - z_1)
\tag{8}
$$

Taking the lateral displacement and the unbalanced rotation of the inertia load into account, the dynamics of dual-cylinder synchronous servo system are thus described by

$$
\begin{aligned}
m\ddot{\tilde{x}}_G &= -K_s \tilde{x}_G + C_{x\omega_1}\omega_1 + C_{x\omega_2}\omega_2 \\
J_1 \ddot{\omega}_1 &= -(F_1 - F_{r1})l_1 \cos \psi + (F_2 - F_{r2})l_2 \cos \psi + \sum_{i=1}^{2} M_{\omega_1 i} \\
J_2 \ddot{\omega}_2 &= -(F_1 - F_{r1})l_1 \sin \psi + (F_2 - F_{r2})l_2 \sin \psi + \sum_{i=1}^{2} M_{\omega_2 i}
\end{aligned}
\tag{9}
$$

where $\tilde{x}_G$ is the displacement of $G$ in relation to the state of equilibrium along the X-axis; $J_1$ and $J_2$ represents the rotational inertia of the mass center $G$; $K_s$ represents lateral displacement stiffness; $F_{r1}$ and $F_{r2}$ represent the friction of the two cylinders severally. The coupling coefficient $C_{x\omega_1}$ and $C_{x\omega_2}$ between the dynamics of the lateral displacement and the dynamics of the rotational angle can be ignored because the construction of the erection mechanism is practically symmetrical [7].

In Equation (9), $M_{\omega_1 i}$ and $M_{\omega_2 i}$ are the moments induced by the lateral forces of dual cylinders which are described as

$$
\begin{aligned}
\sum_{i=1}^{2} M_{\omega_1 i} &\approx C_{x\omega_1} \tilde{x}_G - K_{\omega_1} \omega_1 \\
\sum_{i=1}^{2} M_{\omega_2 i} &\approx C_{x\omega_2} \tilde{x}_G - K_{\omega_2} \omega_2
\end{aligned}
\tag{10}
$$

where $K_{\omega_1}$ and $K_{\omega_2}$ represent the rotational stiffness.

To avoid the effect of excessive internal forces on the stable erection of dual-cylinder synchronous servo system, it is required that the unbalanced rotation angle and lateral moments should be small enough. Noting that the lateral moments in the dynamic Equation (10) is the proportional feedback of the unbalanced rotation angle, thus the synchronous erection requirement can be assured only if the following conditions are satisfied

$$
\begin{aligned}
\left[(F_2 - F_{r2})l_2 - (F_1 - F_{r1})l_1\right] \cos \psi &= 0 \\
\left[(F_2 - F_{r2})l_2 - (F_1 - F_{r1})l_1\right] \sin \psi &= 0
\end{aligned}
\tag{11}
$$

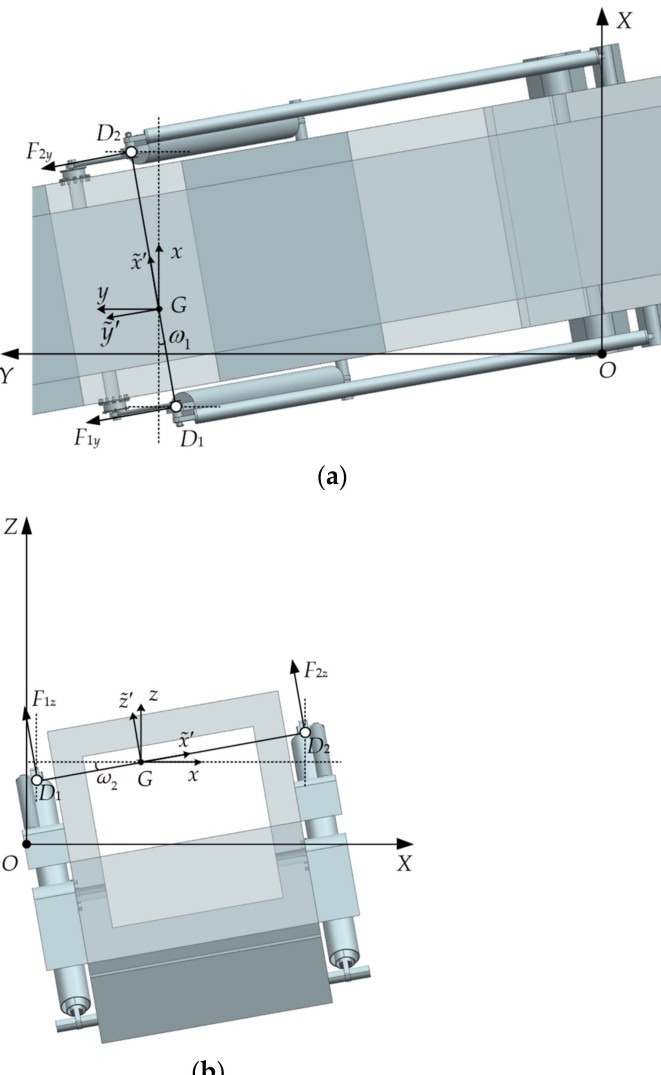

**Figure 6.** Motion analysis diagram of the unbalanced rotation. (**a**) Projection of rotation on plane *Gxy*, (**b**) Projection of rotation on plane *Gxz*.

*2.4. Dynamics of Pump-Controlled Hydraulic Actuator*

The erection mechanism is powered by a typical pump-controlled electro-hydraulic servo system (EHSS). As seen from Figure 7, the telescopic cylinder is driven directly by an axial piston pump, which is powered by a permanent magnet synchronous motor (PMSM). The combination of the three-way direct-acting directional valve and the counterbalance valve is adopted to satisfy the oil replenishment and drain automatically in extension mode and retraction mode of telescopic cylinder. The angular displacement of the erection load and the cylinder chamber pressures can be measured via sensors.

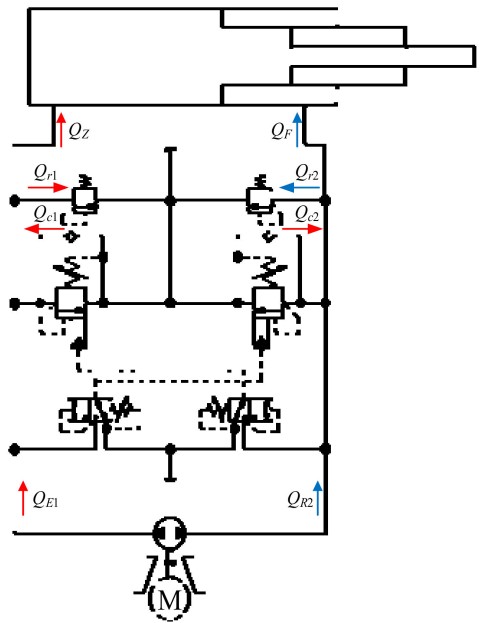

**Figure 7.** Schematic diagram of hydraulic circuit.

The pressure dynamics can be written as [29]

$$\begin{cases} \frac{V_Z}{\beta_e}\dot{P}_Z = Q_Z - A_Z\frac{\partial x_p}{\partial \theta}\dot{\theta} - C_t(P_Z - P_F) \\ \frac{V_F}{\beta_e}\dot{P}_F = Q_F + A_F\frac{\partial x_p}{\partial \theta}\dot{\theta} + C_t(P_Z - P_F) \end{cases} \tag{12}$$

where $V_Z = V_{0Z} + A_Z x_p$ and $V_F = V_{0F} - A_F x_p$ denote the total control volumes of the piston chamber and the rod chamber respectively; $V_{0Z}$ and $V_{0F}$ are the original volumes of the two chambers severally; $\beta_e$ denotes the effective bulk modulus; $P_Z$ and $P_F$ denote the cylinder pressures of the piston chamber and the rod chamber; $Q_Z$ and $Q_F$ denote the flow rates of the two chambers severally; $C_t$ denotes the internal leakage coefficient of the cylinder. Furthermore, the effective areas of the piston chamber and the rod chamber are severally defined as

$$A_Z = \begin{cases} A_{1Z} & x_p \leq l_s \\ A_{2Z} & x_p > l_s \end{cases}, A_F = \begin{cases} A_{1F} & x_p \leq l_s \\ A_{2F} & x_p > l_s \end{cases} \tag{13}$$

where $A_{1Z}$ and $A_{1F}$ are the effective areas of the piston chamber and the rod chamber of the first stage; $A_{2Z}$ and $A_{2F}$ are the effective areas of the piston chamber and the rod chamber of the second stage; $l_s$ is the maximum stroke of the first stage.

Based on the analysis of hydraulic circuit, the flow rates of the two chambers can be described as

$$\begin{cases} Q_Z = Q_{E1} + Q_{c1} - Q_{r1} \\ Q_F = Q_{R2} + Q_{c2} - Q_{r2} \end{cases} \tag{14}$$

where $Q_{E1} = -Q_{R2} = Q$ denotes the supplied and absorbed flow rates of the pump; $Q_{r1}$ and $Q_{r2}$ denote the flow rates through the relief valves, $Q_{r1} = Q_{r2} = 0$, when under normal working conditions; $Q_{c1}$ denotes the drained flow rate when the cylinder retracts, and $Q_{c2}$ denotes the replenished flow rate when the cylinder extends. We define the extension of the cylinder as positive movement, i.e., $\dot{x}_p \geq 0$, then $Q_{c1}$ and $Q_{c2}$ can be written as

$$\begin{cases} Q_{c1}(\dot{x}_p) = \dot{x}_p(A_Z - A_F)sm(-\dot{x}_p) \\ Q_{c2}(\dot{x}_p) = \dot{x}_p(A_Z - A_F)sm(\dot{x}_p) \end{cases} \tag{15}$$

in which $sm(*)$ is defined by

$$sm(*) = \begin{cases} 1 & * \geq 0 \\ 0 & * < 0 \end{cases} \tag{16}$$

For the pump-controlled hydraulic system under study, the flow rates of the pump suction and discharge ports ought to be equal. Thus, the pump flow model can be given as

$$Q = D_p k_\omega u - C_p (P_Z - P_F) \tag{17}$$

where $D_p$ denotes the displacement of the piston pump; $u$ is the control input, which is proportional to the speed of the servo motor via the coefficient $k_\omega$; $C_p$ denotes the pump leakage flow coefficient.

From Equation (4) to Equation (9), the pressure dynamics can be further expressed as

$$\begin{aligned} \frac{V_Z}{\beta_e}\dot{P}_Z &= Q - A_Z \frac{\partial x_p}{\partial \theta}\dot{\theta} - C_t(P_Z - P_F) + \frac{\partial x_p}{\partial \theta}\dot{\theta}(A_Z - A_F)sm(-\frac{\partial x_p}{\partial \theta}\dot{\theta}) \\ \frac{V_F}{\beta_e}\dot{P}_F &= -Q + A_F \frac{\partial x_p}{\partial \theta}\dot{\theta} + C_t(P_Z - P_F) + \frac{\partial x_p}{\partial \theta}\dot{\theta}(A_Z - A_F)sm(\frac{\partial x_p}{\partial \theta}\dot{\theta}) \end{aligned} \tag{18}$$

$x = [x_1, x_2, x_3, x_4]^T = [\theta, \dot{\theta}, A_Z P_{Z1} - A_F P_{F1}, A_Z P_{Z2} - A_F P_{F2}]^T$ is defined as a set of state variables of the erection mechanism; thus, the state-space equations of the system can be described as

$$\begin{aligned} \dot{x}_1 &= x_2 \\ \bar{J}(x_1)\dot{x}_2 &= (x_3 - F_{N1}) + (x_4 - F_{N2}) - f_{21} - Bf_{22} - A_f f_{23} - D(x,t) \\ \dot{x}_3 &= f_{31}\beta_e Q_1 - f_{32}\beta_e - f_{33}C_t\beta_e - f_{34}\beta_e \\ \dot{x}_4 &= f_{41}\beta_e Q_2 - f_{42}\beta_e - f_{43}C_t\beta_e - f_{44}\beta_e \end{aligned} \tag{19}$$

where $\bar{J}(x_1) = J/\tau(x_1)$ and

$$\begin{aligned} f_{21} &= \frac{mgL_8\cos(\beta_2+x_1)}{\tau(x_1)}, f_{22} = \frac{x_2}{\tau(x_1)}, f_{23} = \frac{S_f(x_2)}{\tau(x_1)}, D(x,t) = \frac{d(x,t)}{\tau(x_1)} \\ f_{31} &= \frac{A_Z}{V_Z} + \frac{A_F}{V_F}, f_{32} = (\frac{A_Z^2}{V_Z} + \frac{A_F^2}{V_F})\frac{\partial x_p}{\partial x_1}x_2, f_{33} = (\frac{A_Z}{V_Z} + \frac{A_F}{V_F})(P_{Z1} - P_{F1}) \\ f_{34} &= -\frac{\partial x_p}{\partial x_1}x_2(A_Z - A_F)[\frac{A_Z}{V_Z}sm(-\frac{\partial x_p}{\partial x_1}x_2) - \frac{A_F}{V_F}sm(\frac{\partial x_p}{\partial x_1}x_2)] \\ f_{41} &= \frac{A_Z}{V_Z} + \frac{A_F}{V_F}, f_{42} = (\frac{A_Z^2}{V_Z} + \frac{A_F^2}{V_F})\frac{\partial x_p}{\partial x_1}x_2, f_{33} = (\frac{A_Z}{V_Z} + \frac{A_F}{V_F})(P_{Z2} - P_{F2}) \\ f_{44} &= -\frac{\partial x_p}{\partial x_1}x_2(A_Z - A_F)[\frac{A_Z}{V_Z}sm(-\frac{\partial x_p}{\partial x_1}x_2) - \frac{A_F}{V_F}sm(\frac{\partial x_p}{\partial x_1}x_2)] \end{aligned} \tag{20}$$

Our objective is to synthesize two control inputs for each servo motor to make the erection angle track the reference trajectory and improve the dual-cylinder synchronized motion with decreased internal forces. Before proceeding to the detailed controller design, the following assumption is given.

**Assumption 1.** *The desired motion trajectory $x_{1d} \in C^3$ and is bounded in the actual erection system under conventional operating conditions.*

### 3. Neural Network-Based Synchronous Controller Design

*3.1. Control Model Design*

The controller is designed to make the angular position $x_1(t)$ track the desired erection trajectory $x_{1d}(t)$. The position tracking error $z_1(t)$ is defined as

$$z_1 = x_1 - x_{1d} \tag{21}$$

By using the backstepping method, a switching-function-like quantity is designed as

$$\begin{aligned} z_2 &= \dot{z}_1 + k_1 z_1 = x_2 - x_{2eq}, x_{2eq} \triangleq \dot{x}_{1d} - k_1 z_1 \\ r &= \dot{z}_2 + k_2 z_2, z_3 = x_3 - \alpha_3, z_4 = x_4 - \alpha_4 \end{aligned} \tag{22}$$

where $k_1$ and $k_2$ are positive feedback gains; $x_{2eq}$, $\alpha_3$ and $\alpha_4$ are virtual control laws for $x_2$, $x_3$ and $x_4$, respectively; $r$ is an auxiliary error signal for ease of the subsequent design. Hence, making $z_1$ converging to zero is equivalent to making $z_2$ converge to zero.

In the erection mechanism under research, the values of $\bar{J}(x_1)$, $F_{N1}$ and $F_{N2}$ cannot be measured precisely. Hence, to replace the real values, the nominal values $\bar{J}_0(x_1)$, $F_{0N1}$ and $F_{0N2}$ can be defined as

$$\bar{J}_0(x_1) = \bar{J}(x_1) - \Delta\bar{J}(x_1)$$
$$F_{0N1} = F_{N1} - \Delta F_{N1}, F_{0N2} = F_{N2} - \Delta F_{N2} \tag{23}$$

where $\Delta\bar{J}(x_1)$, $\Delta F_{N1}$ and $\Delta F_{N2}$ are parametric uncertainties. The unmodeled dynamics can be expressed as $D(x_1, x_2, t) = f_1(x_1, x_2) + \Delta(t)$, $\Delta(t)$ is the time-varying disturbance, using (19), (21)–(23), the following expression can be obtained

$$\bar{J}(x_1)r = F_d + S - \Delta(t) + \Delta d + x_3 + x_4 \tag{24}$$

where the auxiliary function $F_d(x_{1d}, \dot{x}_{1d}, \ddot{x}_{1d})$ is defined as

$$F_d \triangleq -\bar{J}_0(x_{1d})\ddot{x}_{1d} - F_{0N1} - F_{0N2} - f_{21}(x_{1d}) - Bf_{22}(x_{1d}, \dot{x}_{1d}) - A_f f_{23}(x_{1d}, \dot{x}_{1d}) \tag{25}$$

and the auxiliary function $S(x_1, x_2, x_{1d}, \dot{x}_{1d}, \ddot{x}_{1d})$ is defined as

$$
\begin{aligned}
S \triangleq\ & \bar{J}(x_1)(k_1\dot{z}_1 + k_2 z_2) - \bar{J}_0(x_1)\ddot{x}_{1d} + \bar{J}_0(x_{1d})\ddot{x}_{1d} - \Delta\bar{J}(x_1)\ddot{x}_{1d} + \Delta\bar{J}(x_{1d})\ddot{x}_{1d} - f_{21}(x_1) \\
& + f_{21}(x_{1d}) - Bf_{22}(x_1, x_2) + Bf_{22}(x_{1d}, \dot{x}_{1d}) - A_f f_{23}(x_1, x_2) + A_f f_{23}(x_{1d}, \dot{x}_{1d}) \\
& - f_1(x_1, x_2) + f_1(x_{1d}, \dot{x}_{1d})
\end{aligned}
\tag{26}
$$

the unknown dynamics $\Delta d$ which can be estimated via an RBF neural network subsequently is given as

$$\Delta d \triangleq -\Delta\bar{J}(x_{1d})\ddot{x}_{1d} - \Delta F_{N1} - \Delta F_{N2} - f_1(x_{1d}, \dot{x}_{1d}) \tag{27}$$

**Assumption 2.** *$f_1(x_1, x_2)$ is a smooth function and the time-varying disturbance $\Delta(t)$ is smooth enough that*

$$|\Delta(t)| \leq \sigma_1, \left|\dot{\Delta}(t)\right| \leq \sigma_2, \left|\ddot{\Delta}(t)\right| \leq \sigma_3 \tag{28}$$

*where $\sigma_1$, $\sigma_2$ and $\sigma_3$ are some unknown positive constants.*

### 3.2. RBF Neural Network Estimation

The characteristics of the RBF neural network, such as simple architecture, accelerating learning speed and avoiding the local minimum issue, which are all crucial factors in real applications, make it suitable for experiment implementation and have universal approximation ability to approximate any nonlinear function. The basic structure of an RBF neural network with five neurons in the hidden layer is illustrated in Figure 8.

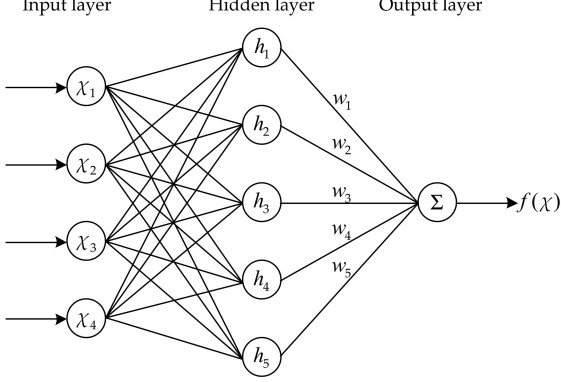

**Figure 8.** Schematic diagram of the RBFNN architecture.

To estimate the unknown dynamics, the RBF neural network is given as follows

$$f(\chi) = \Delta d = W^T h(\chi) + \varepsilon \tag{29}$$

$$h_j = \exp\left(\frac{\|\chi - c_j\|^2}{-2b_j^2}\right) \tag{30}$$

where $\chi = [\chi_1, \chi_2, \chi_3, \chi_4]^T = [1, x_{1d}, \dot{x}_{1d}, \ddot{x}_{1d}]^T$ represents the network input, $h(\chi) = [h_j]^T$ is the Gaussian function utilized in the network, $c_j$ is the center vector of the $j$th network node, $b_j$ is the width of the $j$th node, $W = [w_j]$ represents the ideal weight vector, $\varepsilon$ represents the network approximation error.

In order to determine the number and the center of hidden neurons, the K-means clustering algorithm is employed in the self-organization learning stage [24]. Firstly, initialize the cluster center, and $K$ groups of different samples are selected randomly as the initial center $c_{ji}(0)$, $(j = 1, 2, \ldots, K, i = 1, 2, 3, 4)$. Secondly, compute the distance between the input vector and the cluster centers, and classify them in accordance with the principle of minimum distance, that is, find $j$ to satisfy the following formula

$$j = \operatorname{argmin}\left\|\chi - c_j^n\right\| \tag{31}$$

where $c_j^n$ is the $j$ center of the basis function at $n$ iteration.

Thirdly, use the mean of each cluster as a new cluster center. Then repeat the second and third step until all the samples are finished and the center distribution is no longer changed, the self-learning process can be finished. Finally, the optimal number of hidden neurons is determined by comparing the corresponding estimation errors in the case of different numbers of initial centers utilized in the self-learning process.

Since the ideal weight cannot be obtained accurately, the actual output of the network is

$$\hat{f}(\chi) = \hat{W}^T h(\chi) \tag{32}$$

where $\hat{W}$ denotes the estimation of $W$. Due to the lack of knowledge about the unknown dynamics of the dual-cylinder erection system, the weights and biases are simply initialized so that the NN output is equal to zero, for then the RISE controller keeps the system stable until the RBF neural network begins to learn. However, practical experiments show that it is important to initialize the weights suitably. A good choice is to select $\hat{W}(0)$ equal to zero [30,31].

The weight adaptive law was designed as follows

$$\dot{\hat{W}} = \Gamma h z_2 \tag{33}$$

where $\Gamma$ is a definite weight adaptive velocity matrix.

**Assumption 3.** *Based on Assumption 2, a set of inequalities can be obtained as follows*

$$|\varepsilon| \le \varepsilon_1, |\dot{\varepsilon}| \le \varepsilon_2, |\ddot{\varepsilon}| \le \varepsilon_3 \tag{34}$$

*where $\varepsilon_1$, $\varepsilon_2$ and $\varepsilon_3$ are known positive constants.*

*3.3. Controller Design with Thrust Allocation*

By utilizing (22) and (29)–(32), $r$ can be expanded as the following formula

$$\bar{J}(x_1)r = z_3 + \alpha_3 + z_4 + \alpha_4 + F_d + S - \Delta(t) + \hat{W}^T h + \tilde{W}^T h + \varepsilon \tag{35}$$

where $\widetilde{W} = W - \hat{W}$ denotes the approximation error of ideal weight, and the resultant virtual control law of the total thrust $F_t$ is specified as

$$
\begin{aligned}
F_t &= \alpha_3 + \alpha_4 = F_{ta} + F_{ts}, F_{ts} = F_{ts1} + F_{ts2} \\
F_{ta} &= -F_d - \hat{W}^T h, F_{ts1} = -(k_r + k_s)z_2 \\
F_{ts2} &= -\int_0^t (k_r + k_s)k_2 z_2 + \beta sign(z_2) d\nu
\end{aligned}
\tag{36}
$$

in which $k_r$ is a positive feedback gain and $k_s$ is a positive constant; $F_{ta}$ functions as a model-based feedforward compensation term utilized to acquire a refined model compensation; $F_{ts}$ as a robust control law where $F_{ts1}$ is a linear robust feedback term and $F_{ts2}$ is a RISE-based integral term to remove the impact of approximation error $\varepsilon$ and residual dynamics; $\beta$ is an integral robust feedback gain; and $sign(z_2)$ is a standard signum function with regard to $z_2$.

To avoid excessive internal forces caused by asynchronous motion, the virtual control thrust of each side is synthesized via thrust allocation law as follows

$$
\begin{cases}
[(\alpha_4 - F_{r2})l_2 - (\alpha_3 - F_{r1})l_1] \cos \psi = 0 \\
[(\alpha_4 - F_{r2})l_2 - (\alpha_3 - F_{r1})l_1] \sin \psi = 0 \\
\alpha_3 + \alpha_4 = F_t
\end{cases}
\tag{37}
$$

Since hydraulic forces generated to drive the erection mechanism are relatively much larger than frictions, (37) can be simplified as $\alpha_4 l_2 - \alpha_3 l_1 = 0$, i.e., $\alpha_3/\alpha_4 = l_2/l_1$.

Based on Equations (35) and (36), the time derivative of $r$ can be given as

$$
\bar{J}(x_1)\dot{r} = -\dot{\bar{J}}(x_1)r + \dot{z}_3 + \dot{z}_4 + \dot{F}_{ts} + \dot{S} - \dot{\Delta}(t) + \dot{\widetilde{W}}^T h + \widetilde{W}^T \dot{h} + \dot{\varepsilon}
\tag{38}
$$

in which the time derivative of $F_{ts}$ can be given as

$$
\dot{F}_{ts} = -(k_r + k_s)r - \beta sign(z_2)
\tag{39}
$$

and the Equation (38) can be rewritten as

$$
\bar{J}(x_1)\dot{r} = -\frac{1}{2}\dot{\bar{J}}(x_1)r + \widetilde{N} + N - (k_r + k_s)r - \beta sign(z_2) + \dot{z}_3 + \dot{z}_4
\tag{40}
$$

in which the unmeasurable auxiliary terms $\widetilde{N}(z_1, z_2, r, t)$, $N(\widetilde{W}, \chi, \dot{\chi}, t)$ are defined as

$$
\widetilde{N} \triangleq -\frac{1}{2}\dot{\bar{J}}(x_1)r + \dot{S} + \dot{\widetilde{W}}^T h
\tag{41}
$$

and

$$
\begin{aligned}
N &\triangleq N_B + N_d \\
N_B &\triangleq \widetilde{W}^T \dot{h}, N_d \triangleq -\dot{\Delta}(t) + \dot{\varepsilon}
\end{aligned}
\tag{42}
$$

**Lemma 1.** *Due to continuously differentiable property, the following upper bound of $\widetilde{N}$ can be acquired via the Mean Value Theorem* [32]

$$
\left\| \widetilde{N} \right\| \leq \rho(\|z\|)\|z\|
\tag{43}
$$

*where $z(t) \triangleq [z_1, z_2, r, z_3, z_4]^T$, and the bounding function $\rho(\|z\|) \in \mathbb{R}$ is a positive globally invertible nondecreasing function.*

**Property 1.** *According to Assumptions 2 and 3, we can obtain the inequalities as follows*

$$
\|N_d\| \leq \varsigma_1, \|N_B\| \leq \varsigma_2, \left\| \dot{N}_d \right\| \leq \varsigma_3, \left\| \dot{N}_B \right\| \leq \varsigma_4
\tag{44}
$$

where $\varsigma_1, \varsigma_2, \varsigma_3, \varsigma_4$ are known positive constants.

According to Equations (19) and (22), Equation (40) can be rewritten as

$$
\begin{aligned}
\bar{J}(x_1)\dot{r} &= \dot{x}_3 - \dot{\alpha}_3 + \dot{x}_4 - \dot{\alpha}_4 - \tfrac{1}{2}\dot{\bar{J}}(x_1)r + \tilde{N} + N - (k_r + k_s)r - \beta sign(z_2) \\
&= f_{31}\beta_e Q_1 - f_{32}\beta_e - f_{33}C_t\beta_e - f_{34}\beta_e - \dot{\alpha}_3 + f_{41}\beta_e Q_2 - f_{42}\beta_e - f_{43}C_t\beta_e \\
&\quad - f_{44}\beta_e - \dot{\alpha}_4 - \tfrac{1}{2}\dot{\bar{J}}(x_1)r + \tilde{N} + N - (k_r + k_s)r - \beta sign(z_2)
\end{aligned}
\tag{45}
$$

thus, the control input flow rate $Q_1$ and $Q_2$ can be designed as

$$
\begin{aligned}
Q_1 &= Q_{1a} + Q_{1s}, Q_{1s} = -\frac{k_3 z_3}{f_{31}\beta_e} \\
Q_{1a} &= (f_{31}\beta_e)^{-1}(f_{32}\beta_e + f_{33}C_t\beta_e + f_{34}\beta_e + \dot{\alpha}_3)
\end{aligned}
\tag{46}
$$

$$
\begin{aligned}
Q_2 &= Q_{2a} + Q_{2s}, Q_{2s} = -\frac{k_4 z_4}{f_{41}\beta_e} \\
Q_{2a} &= (f_{41}\beta_e)^{-1}(f_{42}\beta_e + f_{43}C_t\beta_e + f_{44}\beta_e + \dot{\alpha}_4)
\end{aligned}
\tag{47}
$$

in which $Q_{1a}$ and $Q_{2a}$ are model compensation terms, $Q_{1s}$ and $Q_{2s}$ are robust control laws to stabilize the system; $k_3$ and $k_4$ are positive feedback gains. Based on Equations (17), (46) and (47), the actual control inputs are designed as

$$
u_1 = \frac{Q_1 + C_p(P_Z - P_F)}{D_p k_\omega}
\tag{48}
$$

$$
u_2 = \frac{Q_2 + C_p(P_Z - P_F)}{D_p k_\omega}
\tag{49}
$$

Applying the resulting control inputs, Equations (48) and (49), (45) could be changed to

$$
\bar{J}(x_1)\dot{r} = -\frac{1}{2}\dot{\bar{J}}(x_1)r + \tilde{N} + N - (k_r + k_s)r - \beta sign(z_2) - k_3 z_3 - k_4 z_4
\tag{50}
$$

and the dynamics of $z_3$ and $z_4$ are transformed to

$$
\dot{z}_3 = -k_3 z_3, \dot{z}_4 = -k_4 z_4
\tag{51}
$$

The schematic diagram of the proposed synchronous control strategy is depicted in Figure 9.

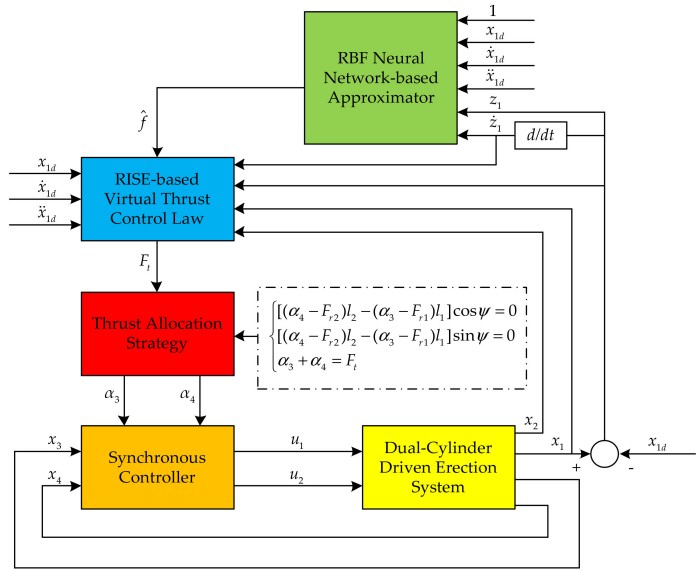

**Figure 9.** Schematic diagram of the proposed synchronous control strategy.

*3.4. Stability Analysis*

**Lemma 2.** *Define an auxiliary function $L(t) \in \mathbb{R}$ as*

$$L(t) \triangleq r[N_d(t) + N_B(t) - \beta sign(z_2)] \tag{52}$$

*From Property 1, there does exist a constant $\beta$ in accordance with the following condition*

$$\beta \geq \varsigma_1 + \varsigma_2 + \frac{\varsigma_3}{k_2} + \frac{\varsigma_4}{k_2} \tag{53}$$

*then based on [22,23], the function $P(t) \in \mathbb{R}$ defined below is always positive*

$$P(t) \triangleq \beta|z_2(0)| - z_2(0)N(0) - \int_0^t L(\nu)d\nu \tag{54}$$

Define $D \subset R^6$ as a domain containing $h(t) = 0$, and $h(t)$ is given as

$$h(t) \triangleq [z(t), \sqrt{P(t)}]^T \tag{55}$$

Define $V_L(y,t)$ as a positive Lyapunov function

$$V_L(y,t) = \frac{1}{2}z_1^2 + \frac{1}{2}z_2^2 + \frac{1}{2}z_3^2 + \frac{1}{2}z_4^2 + \frac{1}{2}\bar{J}(x_1)r^2 + P \tag{56}$$

which satisfies the following inequalities

$$U_1(y) \leq V_L(y,t) \leq U_2(y) \tag{57}$$

In (57), the continuous positive definite functions $U_1(y)$, $U_2(y)$ are defined as

$$U_1(y) \triangleq \lambda_1\|y\|^2, U_2(y) \triangleq \lambda_2\|y\|^2 \tag{58}$$

in which

$$\lambda_1 \triangleq \frac{1}{2}\min\{1, \bar{J}_{\min}\}, \lambda_2 \triangleq \frac{1}{2}\max\{1, \bar{J}_{\max}\}$$

where $\bar{J}_{\max}$ and $\bar{J}_{\min}$ are the maximum and minimum values of function $\bar{J}(x_1)$, severally.

By using Equations (22), (36) and (48)–(50), we can obtain the time derivative of $V_L(y,t)$

$$\begin{aligned}\dot{V}_L(y,t) &= z_1\dot{z}_1 + z_2\dot{z}_2 + z_3\dot{z}_3 + z_4\dot{z}_4 + \bar{J}(x_1)r\dot{r} + \frac{1}{2}\dot{\bar{J}}(x_1)r^2 + \dot{P} \\ &= -k_1z_1^2 + z_1z_2 - k_2z_2^2 + z_2r - (k_r + k_s)r^2 - k_3z_3^2 - k_3rz_3 \\ &\quad -k_4z_4^2 - k_4rz_4 + r\widetilde{N}\end{aligned} \tag{59}$$

By utilizing Equation (43) and choosing appropriate feedback gains $k_1, k_2, k_r, k_3$ and $k_4$, such that the following defined matrix $\Lambda$ is positive

$$\Lambda = \begin{bmatrix} k_1 & -\frac{1}{2} & 0 & 0 & 0 \\ -\frac{1}{2} & k_2 & -\frac{1}{2} & 0 & 0 \\ 0 & -\frac{1}{2} & k_r & \frac{k_3}{2} & \frac{k_4}{2} \\ 0 & 0 & \frac{k_3}{2} & k_3 & 0 \\ 0 & 0 & \frac{k_4}{2} & 0 & k_4 \end{bmatrix} \tag{60}$$

then, Equation (59) can be simplified as

$$\dot{V}_L(y,t) \leq -\kappa_{\min}(\Lambda)\|z\|^2 - (k_s|r|^2 - \rho(\|z\|)|r|\|z\|) \tag{61}$$

in which

$$-(k_s|r|^2 - \rho(\|z\|)|r|\|z\|) = -\left(\sqrt{k_s}|r| - \frac{\rho(\|z\|)\|z\|}{2\sqrt{k_s}}\right)^2 + \frac{\rho(\|z\|)^2\|z\|^2}{4k_s} \leq \frac{\rho(\|z\|)^2\|z\|^2}{4k_s} \quad (62)$$

and $\kappa_{\min}(*)$ denotes the minimum eigenvalue of the argument. Thus, the following inequality can be acquired

$$\dot{V}_L(y,t) \leq -\kappa_{\min}(\Lambda)\|z\|^2 + \frac{\rho(\|z\|)^2\|z\|^2}{4k_s} \leq -U(y) \quad (63)$$

where $U(y) = n\|z\|^2$, for some positive constant $n$, is a continuous positive semi-definite function and defined on the following domain

$$D \triangleq \left\{ y \in R^6 \Big| \|y\| < \rho^{-1}\left(2\sqrt{\kappa_{\min}(\Lambda)k_s}\right) \right\} \quad (64)$$

Based on Equations (56) and (64), it can be known that $V_L \in L_\infty$ in $D$. From Equations (21), (22), (25), (26), (29), (43), (44), (48) and (49), it can be inferred that system signals are bounded. The definitions of $U(h)$ and $z(t)$ prove that $U(y)$ is uniformly continuous in $D$.

Define $\Upsilon \subset D$ as a set as follows

$$\Upsilon \triangleq \left\{ y(t) \subset D \Big| U_2(y(t)) < \lambda_1\left(\rho^{-1}\left(2\sqrt{\kappa_{\min}(\Lambda)k_s}\right)\right)^2 \right\} \quad (65)$$

By invoking [32], it can be stated that

$$n\|z\|^2 \to 0 \ as \ t \to \infty \ \forall \ y(0) \in \Upsilon \quad (66)$$

In addition, according to the definition of $z(t)$, (66) can be used to prove that

$$z_1(t) \to 0 \ as \ t \to \infty \ \forall \ y(0) \in \Upsilon \quad (67)$$

Hence, it can be ensured that all system signals are bounded and a semi-global asymptotic tracking performance can be obtained via the presented control algorithm.

## 4. Simulation Results

The parameters of the dual-cylinder erection mechanism are listed in Table 1. The disturbance was set as $d(t) = 50,000\sin(t) \ N \cdot m$.

**Table 1.** Physical parameters of the dual-cylinder erection mechanism.

| Parameter | Value | Parameter | Value |
|---|---|---|---|
| $m$(kg) | 10,000 | $l_s$(m) | 2.442 |
| $J$(kg·m$^2$) | $2.61 \times 10^5$ | $V_{0Z}$(m$^3$) | $1.075 \times 10^{-4}$ |
| $k_\omega$(rpm/V) | 300 | $V_{0F}$(m$^3$) | $2.83 \times 10^{-2}$ |
| $D_p$(m$^3$/rev) | $1.8 \times 10^{-4}$ | $C_t$(m$^3$/s/Pa) | $4.82 \times 10^{-13}$ |
| $A_{1Z}$(m$^2$) | $2.69 \times 10^{-2}$ | $C_p$(m$^3$/s/Pa) | $2.56 \times 10^{-11}$ |
| $A_{1F}$(m$^2$) | $6.8 \times 10^{-3}$ | $B$(N·m·s/rad) | $8 \times 10^4$ |
| $A_{2Z}$(m$^2$) | $1.43 \times 10^{-2}$ | $A_f$(N·m) | 3500 |
| $A_{2F}$(m$^2$) | $4.8 \times 10^{-3}$ | $\beta_e$(Pa) | $7 \times 10^8$ |

The following four controllers are contrasted to validate the effectiveness of the presented control strategy.

(1) RISE-NN-TAS: This is the neural network-based synchronous controller integrated with the RISE feedback term and a thrust-allocation strategy. The controller gains are: $k_1 = 10$, $k_2 = 10$, $k_3 = 30$, $k_4 = 30$, $k_r = 20$, $\beta = 5$. The RBFNN is designed based

on the K-means clustering algorithm with five neurons in the hidden layer, and the parameters are: $c_j = [-1, -0.5, 0, 0.5, 1]$, $b_j = 10$, $\hat{W}(0) = [0; 0; 0; 0; 0]$. The definite weight adaption velocity matrix is: $\Gamma = diag\{1000, 1000, 1000, 1000, 1000\}$.

(2) RISE-TAS: This is the RISE-based controller with thrust allocation strategy, the designed virtual control law and control inputs are as follows:

$$
\begin{cases}
F_t = -F_d - (k_r + k_s)z_2 - \int_0^t (k_r + k_s)k_2 z_2 + \beta sign(z_2)dv \\
u_1 = (D_p k_\omega)^{-1}[(f_{31}\beta_e)^{-1}(f_{32}\beta_e + f_{33}C_t\beta_e + f_{34}\beta_e + \dot{\alpha}_3 - k_3 z_3) + C_p(P_Z - P_F)] \\
u_2 = (D_p k_\omega)^{-1}[(f_{41}\beta_e)^{-1}(f_{42}\beta_e + f_{43}C_t\beta_e + f_{44}\beta_e + \dot{\alpha}_4 - k_4 z_4) + C_p(P_Z - P_F)]
\end{cases} \quad (68)
$$

The controller gains are corresponding to the RISE-NN-TAS controller for the convenience of comparing the controller performance.

(3) RFC-TAS: This the robust feedback controller introduced in [20] with thrust allocation strategy. In order to ensure the fairness of the comparison, the controller gains $k_1$, $k_2$, $k_3$, $k_4$ are the same as RISE-NN-TAS, and RISE gain coefficients are given by $k_r = 0$, $\beta = 0$.

(4) PID-CC: This is the proportional-integral-derivative controller with cross-coupling compensation, which is widely employed and tested in industrials for synchronous motion control. The P-gain, I-gain and D-gain are chosen as $k_P = 40$, $k_I = 80$, $k_D = 0$, respectively. The cross-coupling gain is tuned as $k_C = 1.5$.

The maximum, average, standard deviation of the errors, denoted as $M_e$, $\mu$ and $\sigma$ severally [21,23], are utilized to assess the efficiency of the aforementioned controllers. These criteria are defined as follows:

(i) Maximal absolute tracking/synchronization error:

$$
M_e = \max_{i=1,\dots,N}\{|e(i)|\} \quad (69)
$$

(ii) Average tracking/synchronization error:

$$
\mu = \frac{1}{N}\sum_{i=1}^{N}|e(i)| \quad (70)
$$

(iii) Standard deviation of the tracking/synchronization error:

$$
\sigma = \sqrt{\frac{1}{N}\sum_{i=1}^{N}(|e(i)| - \mu)^2} \quad (71)
$$

Case 1: The four controllers were first tested for a slow-erection trajectory, which was planned via the constant-power strategy. The maximum angular velocity and acceleration were 0.094 rad/s and 0.033 rad/s$^2$, respectively. In this case, the tracking performance under the proposed controller depicted in Figures 10 and 11 shows the tracking errors of the four controllers. From these simulation results, it can be concluded that the RISE-NN controller attained the best tracking performance in contrast to the other three controllers. Meanwhile, excellent stage-changing and in-position performance were also achieved under the RISE-NN controller. This was due to the compensation for the effects of interstage collision and the approximation of the unknown dynamics acquired from the RBFNN. Figure 12 shows the synchronization errors for the four controllers. It can be seen that both the model-based intelligent controllers with thrust allocation strategy achieved much better synchronization performances than the cross-coupling PID controller, which neglected the effects of unbalanced internal forces. Additionally, a cross-coupling compensation compulsively operated on the dual-cylinder system would result in severe "pull and drag" phenomena and a worse tracking performance. Moreover, the thrust-allocation strategy presented in this paper was easy to implement and capable of eliminating the interference of the dual-cylinder drives. In addition, the estimation for $f$ is provided in Figure 13,

and Figure 14 depicts the control inputs of the presented RISE-NN controller with thrust allocation. As shown in Tables 2 and 3, the superiority of the RBFNN approximation ability and the proposed thrust allocation strategy can be further verified on the basis of listed performance indexes, respectively.

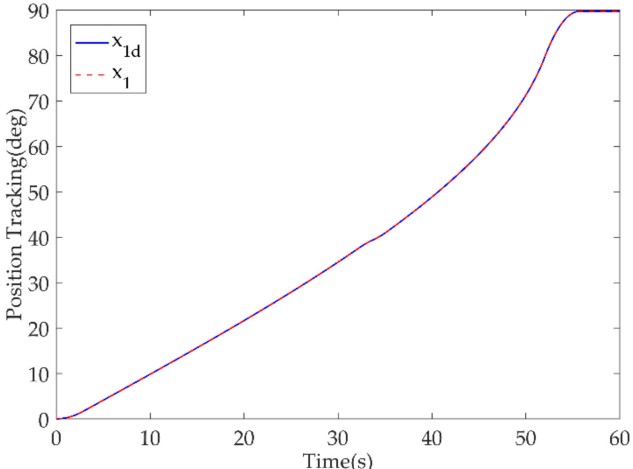

**Figure 10.** Tracking performance under RISE-NN-TAS controller in case 1.

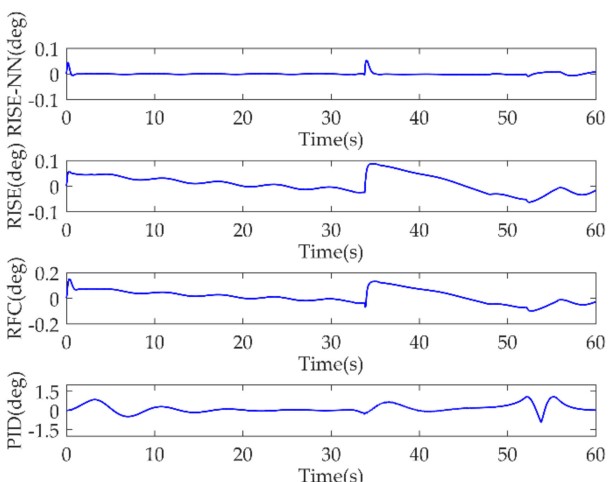

**Figure 11.** Tracking errors of the four controllers in case 1.

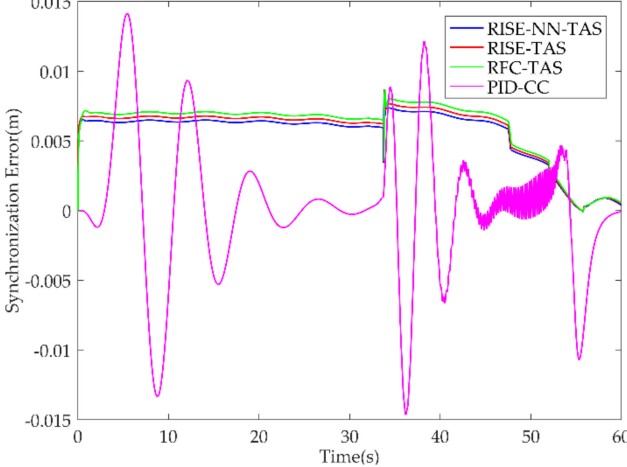

**Figure 12.** Synchronization errors of the four controllers in case 1.

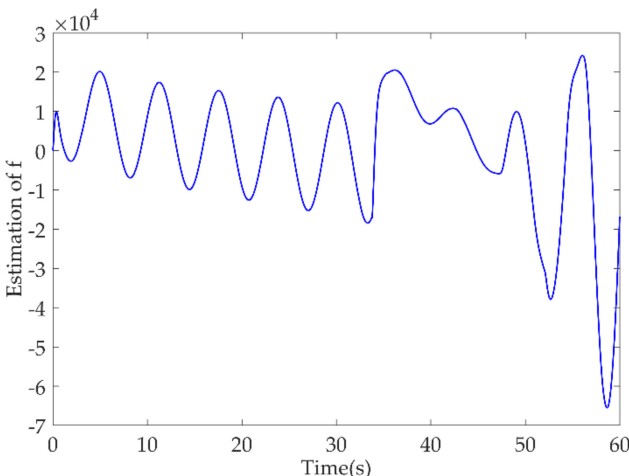

**Figure 13.** RBFNN estimation for $f$.

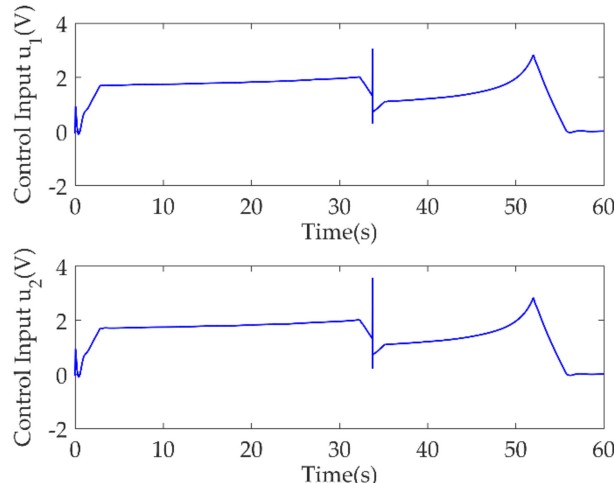

**Figure 14.** Control inputs of RISE-NN-TAS with thrust allocation.

**Table 2.** Tracking performance indexes of the four controllers in case 1.

| Indexes | $M_e$ | $\mu$ | $\sigma$ |
|---|---|---|---|
| RISE-NN | 0.0741 | 0.0042 | 0.0066 |
| RISE | 0.0924 | 0.0338 | 0.0230 |
| RFC | 0.1509 | 0.0433 | 0.0323 |
| PID | 1.0644 | 0.2302 | 0.2552 |

**Table 3.** Synchronization performance indexes of the four controllers in case 1.

| Indexes | $M_e$ | $\mu$ | $\sigma$ |
|---|---|---|---|
| RISE-NN-TAS | 0.0083 | 0.0055 | 0.0018 |
| RISE-TAS | 0.0087 | 0.0058 | 0.0020 |
| RFC-TAS | 0.0089 | 0.0060 | 0.0023 |
| PID-CC | 0.0147 | 0.0033 | 0.0038 |

Case 2: To further validate the effectiveness of the proposed approach, a fast erection trajectory was tested, with the maximum angular velocity 0.133 rad/s and acceleration 0.045 rad/s$^2$. In this case, the inter-stage collision effect raised and led to a stronger transient interference to the system, which could decrease the tracking performance and erection

stability. The tracking performance under the presented controller is shown in Figure 15. Figure 16 shows the tracking errors of the four controllers. Obviously, the stage-changing and in-position performance can still be ensured via the inter-stage collision compensation and the RBFNN approximation of the unknown dynamics. The synchronization errors of the four controllers are shown in Figure 17. It indicates that, with the proposed thrust allocation strategy, the resistance of improving synchronization performance caused by the internal forces was well handled and the "pull and drag" phenomena could be avoided. Furthermore, the performance indexes in case 2 are presented in Tables 4 and 5, which exhibits the superiority of the RBFNN in unknown dynamics approximation and the proposed thrust allocation strategy in solving synchronous control problems coupling with internal forces. Last but not least, virtually identical synchronization performances were ensured via the three different model-based controllers with thrust allocation strategy, which verified the universal applicability for the proposed methods in intelligent synchronous control systems with coupled dynamics.

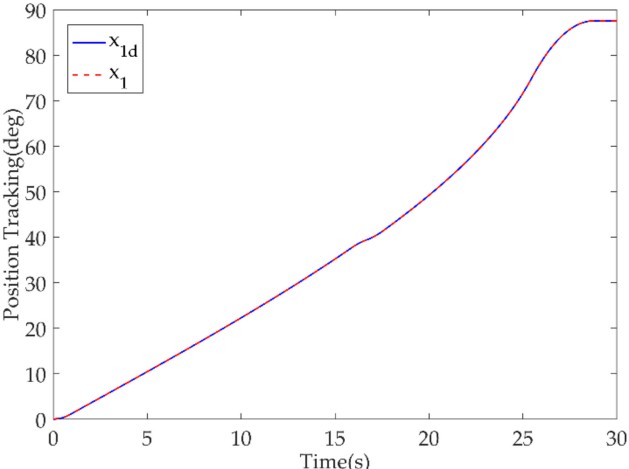

**Figure 15.** Tracking performance under RISE-NN-TAS controller in case 2.

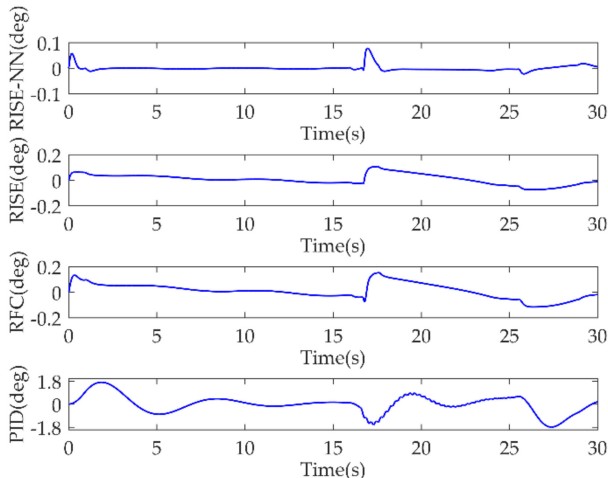

**Figure 16.** Tracking errors of the four controllers in case 2.

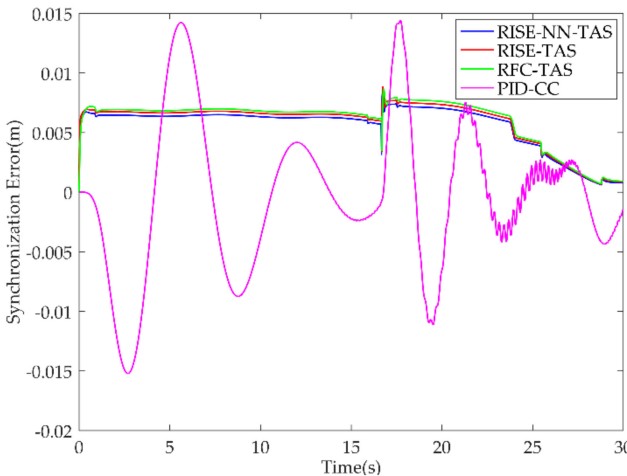

**Figure 17.** Synchronization errors of the four controllers in case 2.

**Table 4.** Tracking performance indexes of the four controllers in case 2.

| Indexes | $M_e$ | $\mu$ | $\sigma$ |
|---------|-------|-------|----------|
| RISE-NN | 0.0760 | 0.0059 | 0.0099 |
| RISE | 0.1026 | 0.0372 | 0.0261 |
| RFC | 0.1525 | 0.0482 | 0.0369 |
| PID | 1.7179 | 0.4394 | 0.4540 |

**Table 5.** Synchronization performance indexes of the four controllers in case 2.

| Indexes | $M_e$ | $\mu$ | $\sigma$ |
|---------|-------|-------|----------|
| RISE-NN-TAS | 0.0084 | 0.0056 | 0.0019 |
| RISE-TAS | 0.0087 | 0.0059 | 0.0022 |
| RFC-TAS | 0.0089 | 0.0062 | 0.0026 |
| PID-CC | 0.0152 | 0.0048 | 0.0042 |

## 5. Conclusions

In this paper, a neural network (NN) based novel synchronous scheme integrated with thrust allocation strategy was proposed for a multi-link erection mechanism driven by a dual-cylinder system. The inter-stage collision dynamics of the telescopic cylinder were established for model compensation, which enhanced stage-changing and in-position tracking performance. An easily designed RBFNN was then utilized for unknown dynamics' approximation and compensation to attain a high-accuracy tracking performance. Meanwhile, a synchronous control strategy with thrust-allocation law was constructed to deal with excessive internal forces caused by unbalanced rotation and lateral moments, which attenuated the impact of the "pull and drag" issue and synchronized the motions of the two cylinders via regulating internal forces. These characteristics determined the intrinsic distinctions from commonly used synchronous controllers for multiple manipulators with decoupled dynamics, which mainly pursue excellent motion synchronization performance under various working conditions. The Lyapunov method was employed to analyze the stability of the designed control algorithm, which indicated that the semi-global asymptotic tracking performance could be attained. Contrasting simulation results were derived to validate the effectiveness of the RBFNN and the proposed synchronous controller with a thrust-allocation strategy. It is worth noting that the implementation of the thrust allocation control strategy depended on the location identification of the center of mass, which might have been tough to be determined precisely. As for future studies, it is vital to conduct a validation experiment on a practical erection platform and introduce on-line adaptation of the thrust allocation factor into synchronous controller design.

**Author Contributions:** Formal analysis, W.Z. and Y.G.; Investigation, W.D., L.L. and J.Y.; Methodology, W.Z., J.Z. and J.Y.; Project administration, W.D. and X.L.; Writing—original draft, W.Z. All authors have read and agreed to the published version of the manuscript.

**Funding:** This research was supported in part by the National Natural Science Foundation of China (51905271, 52075262), in part by the Natural Science Foundation of Jiangsu Province (BK20190459) and in part by the Fundamental Research Funds for the Central Universities (30920041101).

**Conflicts of Interest:** The authors declare no conflict of interest.

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
