# Peer review of "Neural Network-Based Dual-Cylinder Synchronous Control of a Multi-Link Erection Mechanism"

_electronics, doi:10.3390/electronics11162542_

Round 1
Reviewer 1 Report
The paper is very interesting, you show all aspects concerning the mathematical model and the results show a good performance, I only want to know if the RBF is a requisite, and probably Figure 6 can present the scheme in 3D for a better understanding.
Reviewer 2 Report
I am writing in relation to the manuscript with the title of "Neural network based dual-cylinder synchronous control of a multi-link erection mechanism", submitted to the journal of Electronics for peer review.
The manuscript aims to propose a neural-network-based tracking control strategy for a dual-cylinder erection system by synchronizing the motion of two cylinders and decreasing the resulting internal forces.
There are a number of issues related to the manuscript under review that have been highlighted in the comments below:
1) Why using radial basis function neural network (RBFNN), but not other types of neural networks? The authors should have convincing reasons because there are many classes of neural networks in the literature.
2) Although the proposed intelligent controller has been compared with two other classical controllers, there is no comparison between the findings of this study and other similar works available in the literature. Such comparisons can help readers more investigate the tracking capacity of the proposed control strategy.
3) Please provide a schematic diagram of the neural network architecture indicating the layers with their respective weights and biases.
4) Please provide full details about the training process and parameters used in running the neural network model. Explain how the initial connection weights and biases and the number of hidden neurons are determined.
5) Please extend the conclusion section by discussing the disadvantages of the proposed control strategy. Justify your findings with the other similar works in the existing literature.
6) P13/L419: What is meant by 'Theor.8.4'?
7) A series of grammatical errors have been discovered in the text. It is therefore advisable to proofread the whole manuscript carefully.
Round 2
Reviewer 2 Report
The revised version of the manuscript gives neither a reasonable justification for employing the radial basis function network nor any specific method for setting the number of hidden neurons, initial weights and biases. It is necessary to fine-tune the control gains in all the simulation experiments. There is also no block diagram indicating the whole structure of the proposed controller.
As mentioned earlier, the trajectory performance of the proposed control strategy should be compared with the other intelligent controllers implemented on the similar dual-cylinder erection system.On the other hand, it is unlikely a PID controller with hand-tuned parameters can stabilize the given highly nonlinear uncertain plant. So, it cannot be a good choice for control comparison. In addition, it is needed to test the designed controller based on a set of control criteria, e.g. the percentage of overshoot, settling time, and rise time so that the controller performance can be effectively analyzed and compared.
Last but not least, the robustness of the designed controller should be evaluated in terms of external disturbance, energy consumption, and parametric uncertainty.
The main focus of the manuscript is on mathematical modeling rather than control theory. For the above reasons, it seems to me that the contribution of this work to the control theory is vague and unclear.
